# Predominance of Multidrug-Resistant Gram-Negative Bacteria Isolated from Supermarket Retail Seafood in Japan

**DOI:** 10.3390/microorganisms11122935

**Published:** 2023-12-07

**Authors:** Christian Xedzro, Toshi Shimamoto, Tadashi Shimamoto

**Affiliations:** Laboratory of Food Microbiology and Hygiene, Graduate School of Integrated Sciences for Life, Hiroshima University, 1-4-4 Kagamiyama, Higashihiroshima 739-8528, Japan; d211752@hiroshima-u.ac.jp (C.X.); tsima@hiroshima-u.ac.jp (T.S.)

**Keywords:** antimicrobial resistance, retail seafood, plasmid, surveillance, Japan

## Abstract

Reports have documented antimicrobial usage in aquaculture, and the aquatic ecosystem can be considered a genetic storage site for antibiotic-resistant bacteria. This study assessed the prevalence of antimicrobial resistance (AMR) among Gram-negative bacteria recovered from retail seafood in Hiroshima, Japan. A total of 412 bacteria were isolated and screened for the presence of β-lactamases, acquired carbapenemases, and mobile colistin-resistance (*mcr*) genes. Forty-five (10.9%) isolates were dominated by *Morganella* (28%), *Proteus* (22%), *Aeromonas* (14%), *Citrobacter* (8%), and *Escherichia* (8%) and carried AMR genes. The identified AMR genes included those encoded in integrons (19), *aac(6՛)-Ib* (11), *bla*_TEM-1_ (7), *bla*_CTX-M-like_ (12), *bla*_CTX-M-65_ (2), *bla*_SHV-12_ (1), *bla*_SHV-27_ (1), *bla*_OXA-10_ (1), *bla*_OXA-2_ (1), and *mcr* (2). The most common clinical resistances were against ampicillin, colistin, sulfamethoxazole/trimethoprim, tetracycline, and ciprofloxacin. Multidrug resistance (MDR) occurred in 27 (60%) AMR isolates, and multiple antibiotic resistance indices ranged from 0.2 to 0.8. A conjugation experiment showed that 10 of the 11 selected MDR strains harbored conjugable plasmids, although PCR-based replicon typing described seven strains as untypable. IncF replicon was identified in MDR extended-spectrum β-lactamase-producing *Escherichia coli* of the pathogenic B2 phylogroup. Our findings suggest that retail seafood harbors MDR bacteria of human interest that require strict resistance surveillance in the seafood production continuum.

## 1. Introduction

Seafood has great nutritional benefits and economic importance, and serves as a rich source of animal proteins, vitamins, minerals, and essential fatty acids, which makes seafood essential to fight malnutrition, especially in low- and middle-income countries [1,2]. Consequently, seafood consumption has increased over the years, with most seafood products originating from aquaculture-based systems [1]. Aquaculture remains an important food sector and is recognized as a pillar of economic growth [3]. For instance, the total aquaculture production reached 122.6 million tons in 2020 and provided approximately 58.5 million jobs worldwide (FAOSTAT 2022) [4].

Globally, aquaculture production has been associated with negative impacts such as the emergence of fish diseases and loss of aquatic life, compelling farmers to apply antibiotics in intensive culture systems [5]. The increasing demand for seafood products has driven farmers to use antimicrobials as growth enhancers. Several antimicrobials including tetracyclines, quinolones, and penicillin are critical in human medicine and have been approved by the World Health Organization for use in aquaculture [6]. In Japan, antibiotics such as tetracyclines, sulfonamides, macrolides, quinolones, and penicillin are used in seawater aquaculture [7]. These antimicrobials have significant applications in the Asian aquaculture industry for the continuous production of seafood products [8]. Excessive use of antibiotics and other chemotherapeutics in aquaculture has resulted in the emergence and dissemination of antimicrobial resistance across diverse bacterial species in aquatic environments [5,9]. Compared with antimicrobial applications in terrestrial animal production, the use of antimicrobials in aquaculture seems to provide a wider exposure pathway for drug distribution within water, which facilitates the accumulation of antimicrobial residues with potential ecosystem health implications.

Furthermore, aquaculture environments that use antimicrobials may serve as genetic reservoirs for antimicrobial resistance, providing direct pathways for human exposure to antimicrobial-resistant bacteria through the consumption of contaminated seafood [10,11]. In addition, there is indirect or horizontal gene transfer of mobile genetic elements (including plasmids, transposable elements, and superintegrons) from aquatic bacteria to human-related pathogens [8,12]. Antimicrobial resistance (AMR) among seafood-borne bacteria can potentially decrease the effectiveness of antimicrobial therapy in humans and lead to human death, financial burden, and a major reduction in aquaculture production. Estimates suggest that approximately 10 million lives and USD 100 trillion will be lost by 2050 because of AMR if existing control measures to prevent the spread of AMR are compromised [13,14].

A variety of fresh fish and seafood products are served raw in Japanese cuisine. For instance, sashimi—thinly sliced seafood prepared from raw fish—is commonly used in Japanese cuisine [15]. Although processed under special and strict hygienic conditions, the possibility of human exposure to critical-priority bacteria exists. Considering that the use of antimicrobials is a common feature in the Japanese aquaculture industry [16], seafood surveillance for antimicrobial resistance is crucial for monitoring the occurrence and status of resistance traits in seafood.

In 2016, Japan implemented a National Action Plan (NAP) for AMR by outlining specific targets for antimicrobial usage, including those applied in aquaculture (government of Japan, 2020). However, the prevalence of AMR in seafood following NAP implementation remains unknown. In addition, reportedly, there are no updated data on seafood surveillance for the future monitoring of AMR trends. Following the principles outlined in the NAP, strengthening the continuous approach for collecting and reporting AMR data is necessary. Such data evaluated at the local, regional, and national levels play a pivotal role in policymaking and justify the effectiveness of the AMR policy. As antimicrobial resistance surveillance is an important aspect of implementing effective control programs, the current study primarily assessed the prevalence of antimicrobial-resistant foodborne bacteria in retail seafood and examined the conjugal transfer mechanisms that might play a key role in the dissemination of AMR genes.

## 2. Materials and Methods

### 2.1. Sample Collection and Pre-Processing

Fifty seafood samples were collected from 21 retail supermarkets in Hiroshima, Japan, between December 2022 and August 2023. Samples included fish (n = 15), shrimp (n = 13), oysters (n = 4), seaweed (n = 1), squid (n = 16), and clams (n = 1). All samples were transported in thermally insulated bags (at 4 °C) to the laboratory and processed within 24 h of collection. Equal portions from the head, abdominal, and tail regions of fish were pooled and mixed before taking the required portion for analysis. The clam samples were broken using a sterile hammer and scalpel, and meat was collected from two to three specimens.

### 2.2. Microbiological Analyses and Bacterial Isolation

Bacterial isolation was performed as previously described [13]. Briefly, a 25 g portion of each sample was weighed into a sterile stomacher bag and homogenized in 225 mL buffered peptone water (Nissui Pharmaceutical Co., Ltd., Tokyo, Japan) containing 2% NaCl. The resulting homogenate was incubated at 37 °C for 6–8 h and plated on MacConkey agar (Eiken Chemical Co., Ltd., Tochigi, Japan) containing ampicillin (100 µg/mL), meropenem (2 µg/mL), streptomycin (50 µg/mL), or colistin (2 µg/mL). Further, 1000 µL of each suspension was transferred into 9 mL Luria–Bertani (LB) broth (Nacalai Tesque, Inc., Kyoto, Japan) with or without ampicillin and streptomycin. To detect β-lactamase producers, ampicillin broth cultures were plated on MacConkey agar containing ampicillin (100 µg/mL) or meropenem (2 µg/mL). In case of selecting aminoglycoside-resistant strains, streptomycin broth cultures were plated on MacConkey agar containing streptomycin (50 µg/mL). Colistin in broth culture was not performed; instead, antibiotic-free LB cultures were plated on MacConkey agar containing colistin (2 µg/mL) and incubated under experimental conditions. Subsequently, 3–5 morphological distinct colonies were selected and re-cultured on individual antibiotic-containing plates. Finally, 412 Gram-negative bacterial isolates were recovered and purified on LB agar without antibiotics.

### 2.3. Molecular Screening to Detect AMR Determinants

Total genomic DNA was extracted from the boiled lysates. Molecular screening of the 412 isolates was performed to identify extended-spectrum β-lactamase (ESBL)-encoding genes (*bla*_SHV_ and *bla*_CTX-M_), other β-lactamases (*bla*_TEM_, *bla*_OXA-1_, *bla*_OXA-2_, *bla*_OXA-5_, *bla*_OXA-9_, and AmpC), integrons (classes 1, 2, and 3), and aminoglycoside acetyltransferase gene *aac(6՛)-Ib*, as previously described [13,17]. PCR detection of mobile colistin-resistance genes (*mcr-1*–*10*) and acquired carbapenemases (KPC, NDM, OXA-48, VIM, and IMP) was performed [13]. The variable regions of classes 1 and 2 integrons within the integron positive isolates were amplified using primers complementary to the 5′ and 3′ conserved segments of class 1 integrons (5′ CS and 3′ CS) and class 2 integrons (hep74 and hep51). PCR amplification conditions were as follows: initial denaturation at 98 °C for 1 min, 30 cycles of denaturation at 98 °C for 10 s, annealing at 50 °C for 15 s, extension at 68 °C for 2 min, and final extension at 68 °C for 15 s [13]. PCR products were purified using ExoSAP-IT (Thermo Fisher Scientific Baltics UAB, Vilnius, Lithuania) and processed for Sanger sequencing using Eurofins Genomics (Tokyo, Japan). Bacteria were identified by amplifying the full-length (1500 bp) 16S rRNA fragment using 27F (5′-AGAGTTTGATCMTGGCTCAG-3′) and 1492R (5′-CGGYTACCTTGTTACGACTT-3′) primer sets.

### 2.4. Phenotypic Antimicrobial Susceptibility Testing

Susceptibility to 15 antimicrobial agents (from 12 different classes) was investigated using the Kirby–Bauer disk diffusion assay method in compliance with the Clinical and Laboratory Standard Institute (CLSI, 2020) [18]. The antibiotic panel (Eiken Chemical Co., Ltd.) included ampicillin (10 µg), cefotaxime (30 µg), ceftazidime (30 µg), cefepime (30 µg), cefoxitin (30 µg), aztreonam (30 µg), meropenem (10 µg), amikacin (30 µg), kanamycin (30 µg), tetracycline (30 µg), chloramphenicol (30 µg), ciprofloxacin (5 µg), sulfamethoxazole-trimethoprim (23.75/1.25 µg), and fosfomycin (50 µg). Bacterial cultures (single colonies) from each isolate were suspended in sterile normal saline (0.85% NaCl, *w*/*v*) and the optical density was adjusted to 0.5 MacFarland standard. The resulting suspension was surface-spread on Muller–Hinton agar (Eiken Chemical Co., Ltd.) using a sterile cotton swab. Each antibiotic-treated disk was carefully placed on an inoculated agar plate. The plates were incubated at 37 °C for 16–18 h and the results were interpreted based on the interpretative chart supplied with the antimicrobial agent. The minimum inhibitory concentration (MIC) of colistin was determined using the broth microdilution method in cation-adjusted Mueller–Hinton broth (Becton, Dickson and Company, MD, USA), in accordance with the 2021 recommendations of the European Committee on Antimicrobial Susceptibility Testing (EUCAST, 2021) [19]. *E. coli* ATCC 25922 was used as the quality control strain. Isolates displaying intermediate resistance were considered resistant to antimicrobial agents. Multidrug resistance is defined as the resistance to three or more antimicrobial classes [20]. The multiple-antibiotic resistance (MAR) index of the isolates was established as previously described [21]. The MAR index is defined as the ratio of *a* and *b*, where “*a*” is the number of antibiotics to which an isolate was resistant and “*b*” is the number of antibiotics to which the same isolate was exposed. A MAR index higher than 0.2 means that the tested isolates originated from a high-risk contamination source where antibiotics are frequently used.

### 2.5. Detection of E. coli Phylogenetic Group

Quadruplex PCR was performed to detect *E. coli* phylogroups as previously described [22]. PCR amplification targets and techniques for assigning phylogroups were as follows: *arpA*|*chuA*|*yjaA|*TspE4.C2. *E. coli* ATCC 25922 was used as a quality control strain for phylogroup B2, and nuclease-free water served as a negative control.

### 2.6. Conjugation Assay

A conjugation experiment was conducted using a filter-mating conjugation assay as previously described [23]. Exponential-phase lysogeny broth cultures of donor bacteria (11 strains selected based on their genotype and phenotype) and azide-resistant *E. coli* J53 recipient strains were used. Both the donor and recipient strains were mixed at a ratio of 1:9 (100 µL donor: 900 μL recipient) and centrifuged for 3 min at 6000 rpm. The supernatant was removed, and the pellets were resuspended in 200 µL LB broth. The resulting suspension was then plated on a conjugation filter (0.22 µm pore size) on LB agar and incubated for 3–5 h at 37 °C. The filter was removed, placed in 3 mL fresh LB medium, and incubated for 1 h at 37 °C. Transconjugants were selected on LB agar containing 100 µg/mL sodium azide and 100 µg/mL ampicillin. Transconjugants were confirmed using PCR targeting the resistance genes and antimicrobial susceptibility testing (ampicillin alone).

### 2.7. Plasmid Isolation and PCR-Based Replicon Typing (PBRT)

Plasmids were isolated from the 11 selected strains as well as the *E. coli* J53 transconjugants. The alkaline lysis method was used for plasmid preparation as described previously [24], with slight modifications. Briefly, alkaline lysis (lysis buffers I, II, and III) was performed on overnight LB broth cultures for 3–5 min. Supernatants were obtained after centrifugation and mixed with an equal volume of isopropanol. The precipitated nucleic acids were collected and dissolved in 1 mL 70% ethanol. Plasmid DNA was recovered as pellets and finally dissolved in 50 µL of Tris-EDTA (pH 8.0) buffer containing 20 µg/mL DNase-free RNase A. PBRT was then performed to identify the plasmid incompatibility (Inc/rep) groups using 18 primer sets targeting HI1, HI2, I1, X, L/M, N, FIA, FIB, W, Y, P, FIC, A/C, T, FIIs, FrepB, K/B, and B/O, as previously described [25]. The assay was conducted using five multiplex and three simplex PCR assays.

### 2.8. Blast Search and Statistical Analyses

All sequenced data were subjected to a similarity search using the Basic Local Alignment Search Tool (BLAST+2.15.0) available in the National Center for Biotechnology Information (NCBI) database. Phenotypic resistance data were recorded in Microsoft Excel 365 v2310 (Build 16924.20150) and presented as percentages. Statistical analyses were performed using IBM SPSS version 29.0 (SPSS Corp., Armonk, NY, USA). Pearson’s chi-square test or Fisher’s exact test were used to analyze significant differences in the prevalence of AMR determinants among samples or antibiotic resistance among different bacterial species. The results were considered not statistically significant when *p*-values were >0.05.

## 3. Results

### 3.1. Isolation and Identification of Bacterial Species

Among the 50 samples analyzed, 22 (44.0%) were contaminated with Gram-negative bacteria carrying antimicrobial resistance determinants. The contaminated samples included 8/15 (53.3%) fish, 6/13 (46.2%) shrimp, 7/16 (43.8%) squid, 1/4 (25.0%) oyster, and 0/1 (0.0%) seaweed and clams (Figure 1).

A total of 412 Gram-negative bacterial species were isolated. Of these, 45 were found to carry AMR determinants. The most prevalent bacterial species identified to harbor AMR determinants were *Morganella morganii* (28.0%), *Proteus* spp. (22.0%), *Aeromonas* spp. (14.0%), *Citrobacter* spp. (8.0%), and *E. coli* (8.0%) (Table 1). Other identified species were *Enterobacter* spp. (6.0%), *Klebsiella pneumoniae* (2.0%), and *Pseudomonas putida* (2.0%). Fish and squid samples were predominantly contaminated with *Morganella*, *Proteus*, and *Aeromonas*, although none of the squid samples tested positive for *Aeromonas* spp. (Table 1). Four *E. coli* strains belonging to the pathogenic B2, commensal A, and B1 strains were isolated from a single sample of fish, shrimp, oyster, or squid. The results show that seafood is a potential carrier of diverse bacteria carrying AMR genes, which is a risk factor for consumer safety.

### 3.2. Prevalence of β-Lactamase-Encoding Genes and Other AMR Determinants in Seafood Samples

Of the 45 isolates identified as carrying AMR genes, 31.1 and 15.6% carried the *bla*_CTX-M_ and *bla*_TEM_ resistance genotypes, respectively (Table 2). These genes were predominantly found in squid isolates. The *bla*_CTX-M_ type was identified in 25.0% and 33.3% of the fish and shrimp isolates, respectively (*p* > 0.05). Only two isolates carrying *bla*_SHV_ or *bla*_OXA_ were found in fish, shrimp, or squid. The most prevalent AMR determinants identified in this study were genes encoded in integrons cassettes (42.2%). They also confer resistance to trimethoprim, quaternary ammonium compounds, aminoglycosides, and chloramphenicol. The aminoglycoside acetyl transferase gene *aac(6՛)-Ib*, which also confers resistance to aminoglycosides, was prevalent in 11 (24.4%) of the total isolates recovered from seafood. Although none of the isolates tested positive for carbapenemase-encoding genes, we found two (4.4%) that harbored the mobile colistin resistance (*mcr*) gene (Table 2). Generally, no significant differences (*p* > 0.05) were observed in the occurrence rates of AMR determinants among the contaminated samples of different categories.

### 3.3. Antibiogram Profiles of Isolated Gram-Negative Bacteria

The isolates (45) were tested against 15 antimicrobial agents, as illustrated in Figure 2. The minimum inhibitory concentration (MIC) was only determined for colistin. All isolates were resistant to at least one antimicrobial agent. Overall, the highest resistance rate was observed for ampicillin (93.3%), followed by colistin (62.2%), sulfamethoxazole/trimethoprim (48.9%), tetracycline (40.0%), ciprofloxacin (35.6%), and cefoxitin (33.3%) (Figure 2). Meropenem resistance was observed in three (6.7%) isolates, while seven (15.6%) isolates also conferred resistance to cephalosporins. The percentage resistance of the most abundant species is listed in Table 3. Ampicillin resistance (100%) was observed in all isolates except *Proteus* spp. (72.7%), although the differences were not statistically significant (*p* > 0.05). Cephalosporin (third and fourth generation)-resistant phenotypes were found in *M. morganii* 1/14 (7.1%), *P. mirabilis* 1/11 (9.1), *E. coli* 2/4 (50.0%), and *E. cloacae* 3/3 (100%) isolates. Notably, some of these isolates do not carry extended-spectrum β-lactamase-encoding genes responsible for hydrolyzing this drug class. Twenty-five isolates of *Morganella* and *Proteus*, which are known to have some intrinsic polymyxin resistance [26], were all resistant to colistin (MIC; 8 µg/mL to >128 µg/mL). Colistin resistance was also identified in one strain of *C. freundii* and in 3/6 (42.9%) of the *Aeromonas* spp. It is worth mentioning that the *mcr-3.2* and *mcr-10.1* identified in *A. hydrophila* and *E. cloacae*, respectively, were susceptible to colistin (MIC; 1 µg/mL).

### 3.4. MAR Indices among MDR Isolates

Twenty-seven of the forty-five (60.0%) strains exhibited MDR phenotypes (Table 4). MDR is defined as non-susceptibility to three or more antimicrobial classes [20]. MAR index values ranged from 0.20 to 0.8, with the highest MAR index originating from *mcr-10.1*-carrying *E. cloacae* (Table 4). Of the 27 isolates that showed MDR phenotypes, 12 (44.4%) had a MAR index of 0.2–0.27, 11 (40.7%) showed a MAR index of 0.33–0.47, and the remaining isolates had a MAR index of 0.53–0.8, suggesting that the isolates originate from a high-risk contaminated source.

### 3.5. E. coli Phylogroup, Plasmid Transferability, and PBRT

The four *E. coli* strains identified in our study were assigned to phylogroups based on the Clermont classification [22]. Two strains carrying *bla*_CTX-M-65_ were assigned to the pathogenic B2 phylogroup, whereas the other two strains carrying *bla*_TEM-1_ belonged to avirulent A or B1 phylogroups. To assess the involvement of some isolates in the spread of antimicrobial resistance, 11 ampicillin-resistant strains exhibiting MDR phenotypes were selected and used as donors in conjugation experiments. Ampicillin-resistant *mcr-3*-carrying *A. hydrophila* was also included, although conjugation was unsuccessful—possibly because *mcr-3* is located on the chromosome. All other isolates successfully transferred their resistance plasmids and ampicillin resistance traits to *E. coli* J53 (conjugation efficiency: 2.8 × 10^−7^ to 8.6 × 10^−6^). Two or five transconjugants obtained from the two mating pairs were analyzed for resistance gene acquisition. All the selected transconjugants carried the resistance determinants identified in the original strains. Although the strains carried conjugable plasmids, PBRT revealed that most of them carried untypable plasmids. Nonetheless, we found that three *E. coli* isolates harbored the IncFIA, IncFIB, IncFIC, or IncFrepB plasmid replicon types. The same plasmids were identified in *E. coli* J53 transconjugants (Table 5).

## 4. Discussion

Thus, the food chain may be a direct source of antimicrobial-resistant bacteria that affect humans. Various bacterial pathogens have been isolated from seafood [4,27], ready-to-eat (RTE) raw seafood [15,28,29], and seafood processing water [14]. In Japan, the Ministry of Health, Labor, and Welfare has issued and strengthened hygiene management criteria, including a proper refrigeration storage temperature (10 °C or lower) and disinfection procedures for proper sanitary processing of seafood or RTE raw seafood and pickled vegetables [29,30]. However, to the best of our knowledge, data on the incidence of antimicrobial-resistant pathogens in seafood and the potential health risks that might arise from foodborne pathogens are scarce. Therefore, studying the antimicrobial resistance of bacteria isolated from seafood is essential.

In the present study, 45 bacterial isolates obtained from various seafood products were characterized along with their resistance attributes. More than 40% of the seafood products were contaminated with different resistant bacteria belonging to eight genera: *Morganella*, *Proteus*, *Aeromonas*, *Citrobacter*, *Escherichia*, *Enterobacter*, *Klebsiella*, and *Pseudomonas*. These bacterial species have been frequently identified in fish, shrimp, and squid samples and have also been detected in seafood samples in other studies in Japan and France [29,31]. Additionally, several other studies have reported the presence of *Aeromonas*, *Proteus*, *Enterobacter*, *E. coli*, *Vibrio*, *Yersinia*, and *Acinetobacter* in the internal organs and muscles of fish [32,33]. Given that these seafood products are widely consumed in Japan, it would certainly be of high priority to continuously examine them in surveillance programs. Furthermore, the presence of coliform bacteria such as *E. coli*, *Klebsiella*, *Citrobacter*, and *Enterobacter* is an indication of fecal or environmental contamination, although some coliforms can also be found in aquatic environments [34]. The contamination rates of samples differ depending on the geographical location, seasonal factors, method of investigation, and sample size. In view of our small sample size, we are unable to comment on this further, although the contamination rate (44%) found in the current study was consistent with or lower than that reported in previous studies (45–96.7%) [6,15,29]. Periodic surveillance with small sample sizes may be effective for AMR surveillance because it provides fast and frequent data for monitoring AMR trends.

PCR and sequence analyses revealed the presence of various AMR genes in the recovered isolates. The commonly encountered AMR determinants included genes encoded in class 1 integrons cassettes (*aadA1*, *aadA2*, *aadA5*, *aadB-1a*, *dfrA14*, *dfrA12,* or *dfrA17*) (42.2%), CTX-M (*bla*_CTX-M-65_, *bla*_CTX-M-9_, or *bla*_CTX-M-like_) (31.1%), *aac(6՛)-Ib* genes (*aacA4* or *aac(6՛)-Ib3*) (24.4%), as well as TEM genes (*bla*_TEM-1_) (15.6%). The detection of these genes was not surprising, as they have been reported in seafood products in Japan [35] and other countries [36,37,38]. Recently, the molecular characterization of ESBL-producing Enterobacterales with a wide range of drug resistance was retrieved from marine fish [38], chicken, and chicken meat [13,39,40], suggesting a wide distribution of ESBLs. These genes were commonly identified, possibly because of plasmid transfer mechanisms, which play an important role in the rapid dissemination of mobile genetic elements, not only in humans and animals but also within the aquatic environment. Seaweed, oysters, and clams were not proportionally sampled, which reduced the probability of detecting AMR genes. We identified this as a limitation of our study that requires further consideration in future surveillance programs.

Several studies have reported antimicrobial use in aquaculture [5,7,10,41]. The irrational use of antimicrobials has resulted in the emergence of resistant pathogens, including multidrug-resistant strains [5]. Similarly, antibiotic resistance was identified in the isolates recovered in this study. Most of the isolates were resistant to ampicillin and colistin, followed by sulfamethoxazole/trimethoprim, tetracycline, and ciprofloxacin (Figure 2 and Table 3). Resistance to some of these antimicrobials has been reported in isolates obtained from seafood products in Japan [29,35]. These antimicrobials, except colistin, have been frequently used in Japanese culture farms for several years [7]. The common use of such antimicrobials may be a cause of the high resistance conferred by the foodborne bacteria isolated in this study. Colistin resistance was also high and was mainly mediated by *Morganella* and *Proteus*, which have been reported to have an intrinsic resistance to polymyxins, including colistin [26]. We also found three isolates of *Aeromonas* that conferred colistin resistance, which was not surprising because some *Aeromonas* species tend to have inducible resistance to colistin [42]. Cephalosporins are currently among the recommended treatment options for patients with severe multidrug-resistant infections [13,43]. In this context, it is important to mention the third- and fourth-generation cephalosporin resistance observed in this study. Seven (15.6%) isolates that conferred clinical resistance to cefotaxime and ceftazidime also hydrolyzed cefepime, a fourth-generation cephalosporin drug with a wide spectrum of activity against many Gram-negative bacterial pathogens [44] (Figure 2). Of the cephalosporin-resistant isolates, two *E. coli*, one *E. cloacae*, and one *P. mirabilis* strains carried *bla*_CTX-M-65_, *bla*_SHV-12_, and *bla*_CTX-M-9_, respectively, whereas the remaining strains did not carry ESBL genes responsible for hydrolyzing cephalosporins. Surprisingly, the *P. mirabilis* strain harboring the *bla*_CTX-M-9_ genotype was susceptible to ampicillin, suggesting a weak hydrolytic activity or no expression of the *bla*_CTX-9_ gene. MDR was observed in 27 (60%) isolates (Table 4), which was alarming. Considering that the consumption of seafood is a part of Japanese culture, the prevalence of MDR constitutes a potential consumer health risk. Among the MDR isolates, the MAR index ranged from 0.2 to 0.8, with approximately 55% having a MAR index greater than 0.3, indicating that these isolates originated from a high-risk contamination source [21]. The high incidence of MDR can be attributed to multiple resistance mechanisms, such as overexpression of multidrug efflux pumps and effective enzymatic hydrolysis of drugs [45]. To prevent the persistence of antimicrobial resistance, possible strategies such as limiting the use of antibiotics in farming practices can help reduce the selective pressure on resistant microorganisms. Additionally, proper sanitation processes in food production can prevent the risk of cross-contamination and the spread of AMR along the food supply chain.

Specific plasmids associated with resistance gene dissemination have been identified in both clinical settings and food chains. The frequent detection of some plasmid types indicates their role in spreading drug resistance traits, especially among *Enterobacteriaceae*. In the present study, we investigated the transferability of plasmids associated with the spread of resistance traits. Even though seven out of the eleven selected strains with conjugable plasmids were negative (untypable) for all our target replicons, we detected IncF replicons in three *E. coli* strains. It is possible that the untypable plasmids were divergent or novel, and PBRT could not detect them because it targets classic incompatibility groups [46]. The detection of IncF replicons in our study is consistent with a previous report that IncF plasmids are common in *Enterobacteriaceae*, especially *E. coli* [46]. Evidence suggests that IncF is the most commonly described plasmid type identified in animals and humans, with the most frequently detected resistance genes being ESBLs [47], suggesting that the plasmids identified in seafood may have originated from human or animal sources. We identified IncFIA, IncFIB, IncFIC, and IncFrepB in ESBL-producing *E. coli* strains belonging to pathogenic B2 or commensal A phylogroups. Pathogenic B2 and commensal A phylogroups were recently reported in a collection of MDR ESBL-producing *E. coli* [13]. In a recent report, IncF-positive commensal MDR *E. coli* strains were recovered from clams and sediments in Italy [48]. In Japan, clinical isolates of MDR *E. coli* were found to carry IncFrepB, IncFIA, and IncFIB replicons [49], further suggesting that contamination of seafood products originated from human or animal sources. The presence of these conjugative plasmids contributes to the spread of MDR traits in seafood.

## 5. Conclusions

In summary, our study revealed the presence of AMR determinants and resistant bacteria in supermarket retail seafood samples collected from Hiroshima, Japan. The multiple antibiotic-resistant phenotypes and indices evaluated in this study suggest an alarming incidence of AMR in seafood, which could pose serious health risks to consumers. Our findings also indicate, to some extent, a high propensity for horizontal gene transfer of mobile genetic elements in seafood and provide baseline information on AMR in the current study region. Therefore, continuous surveillance programs are needed to monitor resistance patterns in seafood. In future studies, we will consider larger sample sizes and/or samples from aquaculture farms to draw definitive conclusions on the status of AMR/MDR originating from seafood in Japan.

## Figures and Tables

**Figure 1 microorganisms-11-02935-f001:**
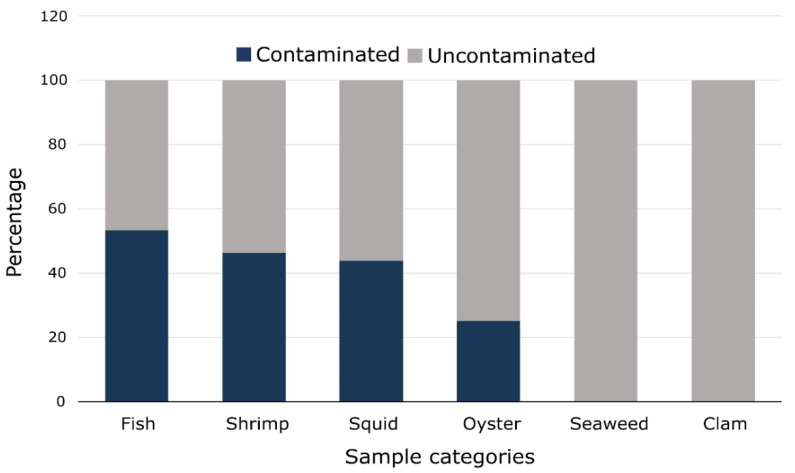
Bar graph showing the percentage of seafood samples contaminated in each sample category.

**Figure 2 microorganisms-11-02935-f002:**
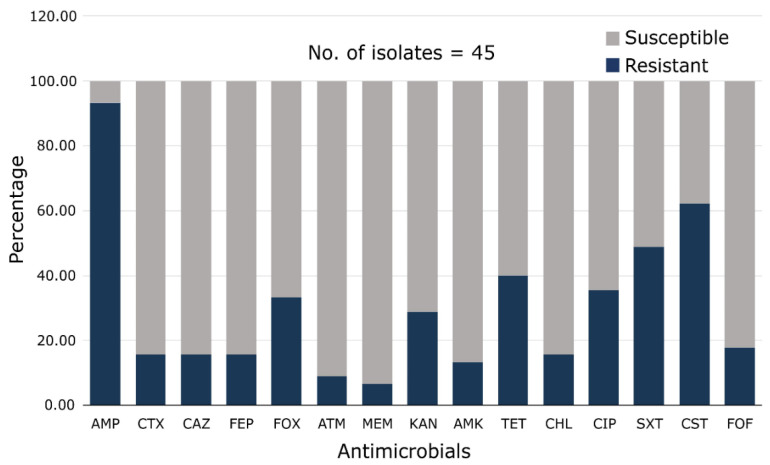
Overall antimicrobial resistance profiles of Gram-negative bacteria isolated from retail seafood. AMP: ampicillin, CTX: cefotaxime, CAZ: ceftazidime, FEP: cefepime, FOX: cefoxitin, ATM: aztreonam, MEM: meropenem, KAN: kanamycin, AMK: amikacin, TET: tetracycline, CHL: chloramphenicol, CIP: ciprofloxacin, SXT: sulfamethoxazole/trimethoprim, CST: colistin, FOF: fosfomycin.

**Table 1 microorganisms-11-02935-t001:** Occurrence of different Gram-negative bacteria isolated from retail seafood in Hiroshima.

Bacterial Species	Prevalence (%)						
	Fish (n = 15)	Shrimp (n = 13)	Oyster (n = 4)	Seaweed (n = 1)	Squid (n = 16)	Clam (n = 1)	Total (N = 50)
*Morganella morganii*	5 (33.3)	3 (23.1)	0 (0.0)	0 (0.0)	6 (37.5)	0 (0.0)	14 (28.0)
*Proteus* spp.	4 (26.7)	2 (15.4)	0 (0.0)	0 (0.0)	5 (31.3)	0 (0.0)	11 (22.0)
*Aeromonas* spp.	4 (26.7)	3 (23.1)	0 (0.0)	0 (0.0)	0 (0.0)	0 (0.0)	7 (14.0)
*Citrobacter* spp.	1 (6.7)	2 (15.4)	0 (0.0)	0 (0.0)	1 (6.3)	0 (0.0)	4 (8.0)
*Escherichia coli*	1 (6.7)	1 (7.7)	1 (25.0)	0 (0.0)	1 (6.3)	0 (0.0)	4 (8.0)
*Enterobacter* spp.	1 (6.7)	0 (0.0)	0 (0.0)	0 (0.0)	2 (12.5)	0 (0.0)	3 (6.0)
*Klebsiella* *pneumoniae*	0 (0.0)	1 (7.7)	0 (0.0)	0 (0.0)	0 (0.0)	0 (0.0)	1 (2.0)
*Pseudomonas putida*	0 (0.0)	0 (0.0)	0 (0.0)	0 (0.0)	1 (6.3)	0 (0.0)	1 (2.0)

N: total number of samples, n: number of samples for each seafood category.

**Table 2 microorganisms-11-02935-t002:** Distribution of AMR genes among retail seafood samples.

Sample Type	No. of Isolates	* β-Lactamase Types (%)				* Others (%)			
		*bla* _SHV_	*bla* _CTX-M_	*bla* _TEM_	*bla* _OXA_	*aac(6՛)-Ib*	Integrons	*mcr*	Carbapenemase
Fish	16	1 (6.3)	4 (25.0)	1 (6.3)	1 (6.3)	5 (31.3)	7 (43.8)	1 (6.3)	0 (0.0)
Shrimp	12	1 (8.3)	4 (33.3)	2 (16.7)	0 (0.0)	1 (8.3)	4 (33.3)	1 (8.3)	0 (0.0)
Oyster	1	0 (0.0)	1 (100)	0 (0.0)	0 (0.0)	0 (0.0)	0 (0.0)	0 (0.0)	0 (0.0)
Seaweed	0	0 (0.0)	0 (0.0)	0 (0.0)	0 (0.0)	0 (0.0)	0 (0.0)	0 (0.0)	0 (0.0)
Squid	16	0 (0.0)	5 (31.3)	4 (25.0)	1 (6.3)	5 (31.3)	8 (50.0)	0 (0.0)	0 (0.0)
Clam	0	0 (0.0)	0 (0.0)	0 (0.0)	0 (0.0)	0 (0.0)	0 (0.0)	0 (0.0)	0 (0.0)
Total (%)	45	2 (4.4)	14 (31.1)	7 (15.6)	2 (4.4)	11 (24.4)	19 (42.2)	2 (4.4)	0 (0.0)

* Indicates no significant difference (*p* > 0.05) in the occurrence of AMR genes among samples.

**Table 3 microorganisms-11-02935-t003:** Percent resistance of the most abundant Gram-negative bacteria against various antimicrobials.

Species	No. of Isolates	* AMP	CTX	CAZ	FEP	FOX	* ATM	* MEM	KAN	AMK	* TET	* CHL	CIP	* SXT	CST	FOF
*Morganella*	14	100	7.1	7.1	7.1	21.4	7.1	0.0	21.4	0.0	42.9	14.3	21.4	50.0	100	35.7
*Proteus*	11	72.7	9.1	9.1	9.1	0.0	9.1	9.1	18.2	9.1	45.5	18.2	36.4	36.4	100	0.0
*Aeromonas*	7	100	0.0	0.0	0.0	57.1	0.0	0.0	14.3	14.3	14.3	0.0	0.0	28.6	42.9	0.0
*Citrobacter*	4	100	0.0	0.0	0.0	100	0.0	0.0	25.0	0.0	50.0	25.0	50.0	100	25.0	0.0
*Escherichia*	4	100	50.0	50.0	50.0	0.0	25.0	0.0	75.0	50.0	50.0	25.0	75.0	25.0	0.0	0.0
*Enterobacter*	3	100	100	100	100	100	33.3	0.0	100	66.7	33.3	0.0	100	100	0.0	0.0

* Indicates no significant difference (*p* > 0.05) in antibiotic resistance among bacterial species. AMP: ampicillin, CTX: cefotaxime, CAZ: ceftazidime, FEP: cefepime, FOX: cefoxitin, ATM: aztreonam, MEM: meropenem, KAN: kanamycin, AMK: amikacin, TET: tetracycline, CHL: chloramphenicol, CIP: ciprofloxacin, SXT: sulfamethoxazole/trimethoprim, CST: colistin, FOF: fosfomycin.

**Table 4 microorganisms-11-02935-t004:** Multidrug resistance phenotypes and multiple antibiotic resistance (MAR) indices of Gram-negative bacteria isolated from retail seafood.

Isolate	Sample	Identification	Genotypes	MDR Profiles	MAR Index
D2-C042	Squid	*Proteus mirabilis*	*bla*_CTX-M-9_, *bla*_OXA-2_	CTX, CAZ, FEP, ATM, KAN, AMK, CIP, SXT, CST	0.60
B2-S021	Shrimp	*Klebsiella pneumoniae*	*bla* _SHV-27_	AMP, MEM, CIP, FOF	0.27
B2-S020	Shrimp	*Citrobacter freundii*	*bla*_TEM-1B_, *dfrA14*	AMP, FOX, CIP, SXT	0.27
D2-S025	Squid	*Proteus vulgaris*	*dfrA1*-*catB2*-*aadA1*	AMP, MEM, TET, CIP, SXT, CST	0.40
A2-A044	Oyster	*Escherichia coli*	*bla* _CTX-M-65_	AMP, CTX, CAZ, FEP, ATM, KAN, CIP	0.47
A3-A049	Fish	*Proteus mirabilis*	*bla* _CTX-M-like_	AMP, TET, CST	0.20
A3-A053	Fish	*Escherichia coli*	*bla* _CTX-M-65_	AMP, CTX, CAZ, FEP, KAN, AMK, CIP	0.47
C3-A056	Shrimp	*Escherichia coli*	*bla* _TEM-1_	AMP, TET, CHL, CIP, SXT	0.33
B3-A054	Squid	*Morganella morganii*	Class 1 integron	AMP, FOX, CST, FOF	0.27
B3-M090	Squid	*Pseudomonas putida*	Class 1 integron	AMP, MEM, CHL, SXT, FOF	0.33
D6-A190	Shrimp	*Aeromonas hydrophila*	*bla* _CTX-M-like_	AMP, FOX, CST	0.20
C6-S206	Fish	*Citrobacter freundii*	*bla*_OXA-10_, *aadA1-bla*_CARB-2_	AMP, FOX TET, CHL, CIP, SXT, CST	0.47
C6-S212	Fish	*Morganella morganii*	*aac(6՛). Ib3, aadA-aadB, bla* _DHA_	AMP, FOX, TET, CIP, SXT, CST	0.40
C6-A180	Fish	*Proteus cibarius*	*bla* _CARB-2_	AMP, TET, CHL, CIP, SXT, CST	0.40
B7-S201	Fish	*Aeromonas allosaccharophila*	*aadA2*, *bla*_FOX_	AMP, FOX, SXT	0.20
B7-S203	Fish	*Aeromonas sobria*	*aadA1-catB8*	AMP, FOX, SXT, CST	0.27
A9-A238	Squid	*Enterobacter cloacae*	*bla*_TEM-1_, *dfrA17*-*aadA5*	AMP, CTX, CAZ, FEP, FOX, KAN, AMK, CIP, SXT	0.60
B14-A346	Squid	*Citrobacter braakii*	*bla* _TEM-1_	AMP, FOX, KAN, TET, SXT	0.33
B12-S377	Fish	*Enterobacter cloacae*	*bla*_TEM-1_, *bla*_SHV-12_, *mcr-10.1*	AMP, CTX, CAZ, FEP, FOX, ATM, KAN, AMK, TET, CIP, SXT, FOF	0.80
D7-C395	Squid	*Escherichia coli*	*bla* _TEM-1_	AMP, KAN, AMK, TET	0.27
B14-C405	Squid	*Proteus alimentorum*	*bla* _CTX-M-like_	AMP, KAN, CST	0.20
B10-C300	Squid	*Morganella morganii*	*aacA4*	AMP, CIP, CST	0.20
A8-C303	Fish	*Morganella morganii*	*aacA4*	AMP, CST, FOF	0.20
A10-C310	Fish	*Morganella morganii*	*aacA4*	AMP, CST, FOF	0.20
C10-C315	Squid	*Morganella morganii*	*aacA4*	AMP, CTX, CAZ, FEP, FOX, ATM, SXT, CST	0.53
B14-S360	Squid	*Morganella morganii*	*aacA4*, *dfrA12*-*aadA2*	AMP, KAN, TET, CIP, SXT CST	0.40
B13-M388	Shrimp	*Morganella morganii*	*aadA1*-*aadB-1a*-*cmlA6*	AMP, TET, CHL, SXT, CST, FOF	0.40

**Table 5 microorganisms-11-02935-t005:** Results of conjugation and plasmid replicon typing of selected MDR bacteria isolated from retail seafood.

Isolate	Species	Genotypes	Inc Groups	Plasmid Transfer	Inc Groups (TC)	Genotype (TC) *	AMP-R (TC)
A2-A044	*E. coli*	*bla* _CTX-M-65_	IncFIA, IncFIB	Transferred	IncFIA	*bla* _CTX-M_	Resistant
A3-A053	*E. coli*	*bla* _CTX-M-65_	IncFIB, IncFIC, IncFrepB	Transferred	IncFIB, IncFIC, IncFrepB	*bla* _CTX-M_	Resistant
C3-A056	*E. coli*	*bla* _TEM-1_	IncFIA, IncFIB	Transferred	IncFIA, IncFIB	*bla* _TEM_	Resistant
C6206	*C. freundii*	*bla*_OXA-10_, *aadA1*-*bla*_CARB-2_	Untypable	Transferred	Untypable	Class 1 integron, *bla*_OXA-5_	Resistant
D6-A192	*A. hydrophila*	*mcr-3.2*	ND	Not transferred	ND	ND	-
C6-A180	*P. cibarius*	*bla* _CARB-2_	Untypable	Transferred	Untypable	Class 1 integron	Resistant
B13-M388	*M. morganii*	*aadA1*-*aadB*-*1a*-*cmlA6*	Untypable	Transferred	Untypable	ND	Resistant
C6-S212	*M. morganii*	*aac(6՛)-Ib3*, *aadA*-*aadB*, *bla*_DHA_	Untypable	Transferred	Untypable	*aac(6* *՛* *)-Ib*	Resistant
B12-S377	*E. cloacae*	*bla*_TEM-1_, *bla*_SHV-12_, *mcr-10.1*	Untypable	Transferred	Untypable	*bla*_CTX-M_, *bla*_SHV_	Resistant
C10-C315	*M. morganii*	*aacA4*	Untypable	Transferred	Untypable	*aac(6* *՛* *)-Ib*	Resistant
A9-A238	*E. cloacae*	*bla*_TEM-1_, *aadA5*-*dfrA17*	Untypable	Transferred	Untypable	*bla* _TEM_	Resistant

TC: transconjugant. AMP-R: ampicillin resistance. ND: not determined. * Resistance genes in the transconjugants were only confirmed by PCR. Note: *bla*_CARB-2_ is a β-lactamase gene, which was found in integron cassettes.

## Data Availability

All data are provided within the article.

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
