# Peer review of "Predominance of Multidrug-Resistant Gram-Negative Bacteria Isolated from Supermarket Retail Seafood in Japan"

_microorganisms, 2023, doi:10.3390/microorganisms11122935_

Round 1

Reviewer 1 Report

Comments and Suggestions for Authors

The authors of the current paper elaborately analyzed the prevalence of multidrug-resistant bacteria in retail seafood sampled in some supermarkets located in Hiroshima (Japan) and, furthermore, examined some of the mechanisms involved in the diffusion of antimicrobial-resistance genes.

The investigation included the bacterial isolation of the microorganisms from the collected samples, the detection of AMR determinants using molecular techniques and the evaluation of the antimicrobial susceptibility as well as some in-depth analyses about the diffusion of the antimicrobial-resistance.

Although the number of samples analysed is quite low and heterogeneous, this study confirms the presence of AMR determinants and resistant bacteria in retail seafood sampled from the supermarkets in Hiroshima and suggests the need of creating continuous surveillance programs to monitor resistance patterns in seafood. 

However, there are some issues that need to be addressed before publication:

- line 187: please replace “An MAR index…” with “A MAR index…”.

- Statistical analysis: authors should change the sentence "The results were considered statistically insignificant when p-values were >0.05" to "The results were considered statistically insignificant when p-values were < 0.05"

- lines 243 and 38: please correct “7/16 (43.8%) oysters” with “7/16 (43.8%) squids”.

- lines 250-257: please format in italic the genus and species of each bacterium.

- lines 296-308: please format in italic the genus and species of each bacterium.

- line 156: please use italic for “E. cloacae”.

- line 340: please use italic for “E. coli”.

- lines 347-348: please use italic for “A. hydrophila”.

- lines 351, 357 and 359: please use italic for “E. coli”.

- line 413: since “Enterobacterales” is an order, italic is not required.

- line 430: please remove “isolated” or replace it with a different verb such as, for example, “obtained” or “coming”.

- line 472: please replace “could not detected” with “could not detect them”

- lines 531-532: please replace “Accessed on October 10th, 2023. [URL]” with “Available online: URL (accessed on 10th October 2023).

- lines 551-554: please add page range.

- lines 559-562: please add page range.

- lines 572-573: please add URL and the date of access

- line 574: please and add the date of access.

- line 626: please replace “2020.” With “2020,”.

- line 642: please add page range.

Comments on the Quality of English Language

I suggest a revision of the English language to improve the reading flow

Author Response

The author's reply to the review report is attached.

Tadashi Shimamoto

Reviewer 2 Report

Comments and Suggestions for Authors

The work is focused on the determination of the predominance of multi-drug-resistant gram-negative microorganisms in seafood in Japan. The authors analyzed an interesting amount of samples and several analyses were carried out to confirm the antimicrobial resistance. I have some minor comments or questions I would appreciate the authors clarifying.

Line 174: MIC was only determined for colistin? This fact should be explained in the results section.

Line 190: Why only the detection of phylogroups was applied to E. coli isolates? 

Line 248: in the abstract, the authors mention 412 isolates where 45 presented AMR genes. This fact should also be commented in the results section

Line 249-261: Please correct the microorganism names in italics along the text.

Table 2: try to improve the presentation of the data in the table, some data appear divided into two lines  and it isn't very clear

Line 297:  I think that the data 72.7 % is not correct since most of the isolates were resistant to ampicillin.

I think that it would be great if some strategies to overcome the presence of resistant microorganisms could be included in the text, especially considering that some isolates presented resistance against third and fourth-generation antibiotics.

Author Response

(The authors gave the same response as above.)
